# The First Report of a Water Mite *Unionicola* (Trombidiformes: Unionicolidae) Infection in *Filopaludina* spp. (Gastropoda: Viviparidae) from Thailand with a Description of a New Species of *Unionicola* (*Polyatax*) *kasetbangkhenensis* sp. nov. [note 1]

**DOI:** 10.3390/ani14111627

**Published:** 2024-05-30

**Authors:** Phuphitchan Rachprakhon, Poramad Trivalairat, Krittiya Trivalairat, Pichit Wiroonpan, Watchariya Purivirojkul

**Affiliations:** 1Department of Zoology, Faculty of Science, Kasetsart University, Bangkok 10900, Thailand; phuphitchan.works@gmail.com; 2Princess Agrarajakumari College of Nursing, Chulabhorn Royal Academy, Bangkok 10210, Thailand; 3Animal Systematics and Ecology Speciality Research Unit (ASESRU), Department of Zoology, Faculty of Science, Kasetsart University, Bangkok 10900, Thailand; chk.krittiya@gmail.com; 4Department of Parasitology and Entomology, Faculty of Public Health, Mahidol University, Bangkok 10400, Thailand; pichit.wir@mahidol.ac.th; 5Biodiversity Center Kasetsart University (BDCKU), Bangkok 10900, Thailand

**Keywords:** water mite, *Unionicola*, *Polyatax*, *Unionicola* (*Polyatax*) *kasetbangkhenensis*, gastropods, *Filopaludina*

## Abstract

**Simple Summary:**

Simple Summary: Water mites are a significant population of Arachnida in aquatic ecosystems, with over 7500 species described worldwide. The genus *Unionicola*, belonging to the family Unionicolidae, is known for its parasitic behavior, often consuming the mucus and tissue of its molluscan hosts. While most species in this genus parasitize mussels, some have been found in gastropods. In our study, we discovered water mites infesting two species of gastropods: *Filopaludina sumatrensis polygramma* and *F. martensi martensi*. Through a comparison with known species, we identified this mite as a new species within the genus *Unionicola*, subgenus *Polyatax*, and named it “*Unionicola* (*Polyatax*) *kasetbangkhenensis* sp. nov.”.

**Abstract:**

Two species of gastropods, *Filopaludina sumatrensis polygramma* and *F. martensi martensi*, were found infested with water mites of the genus *Unionicola* in Bangkok and Nonthaburi provinces, Thailand. Morphological studies on these water mites, based on the characteristics of their genital acetabular and female genital fields, identified them as a new member of subgenus *Polyatax*. *Unionicola* (*Polyatax*) *kasetbangkhenensis* sp. nov. was named after the first location where this parasitic water mite was discovered. This species is distinguished from others by the pattern of their female genital field, with their anterior acetabular plates each bearing two acetabula and a short thick spine on the inner margin. Additionally, it differs from other species by the structure of the spines in its pedipalps and legs, as well as in the shapes of its coxal plates.

## 1. Introduction

Water mites (*Acariformes*: *Hydrachnidia*) are a significant population of Arachnida in aquatic ecosystems, with over 7500 species described worldwide [1]. Most aquatic habitats where water mites live contain freshwater, while most members of the family Pontarachnidae are found in marine environments [2]. Most water mites are predators [3], but some groups of water mites have been reported as parasites of other aquatic animals, such as unionicolid mites (Family Unionicolidae), which are associated with freshwater sponge, mussels, and snails [4,5]. This family is distributed worldwide and consists of five genera with at least 40 described species [6]. Significantly, the genus *Unionicola* is frequently documented as being parasitic, ingesting the mucus and tissue of its molluscan hosts [7,8]. In Thailand, six species in four subgenera have been reported: *Unionicola* (*Fulleratax*) *robacki* Vidrine, 1984; *U*. (*Fulleratax*) *davisi* Vidrine, 1984; *U*. (*Parasitatax*) *brandti* Vidrine, 1985; *U*. (*Pentatax*) *affinis* Piersig 1906; *U*. (*Pentatax*) *thaiensis* Vidrine, 1985; and *U*. (*Polyatax*) *heardi* Vidrine, 1985. The hosts of *U.* (*Fulleratax*) *robacki* are mussel *Hyriopsis bialatus*, *Hy. myersiana*, and *Pilsbryoconcha exilis compressa.* The host of *U*. (*Fulleratax*) *davisi* is mussel *Pi. e. compressa.* The hosts of *U*. (*Parasitatax*) *brandti* are mussel *Uniandra contradens*, *Pseudodon vondembuschianus ellipticus*, *Physunio eximius*, *Trapezoideus exolescens exolescens*, and *Ensidens ingallsianus*. The hosts of *U*. (*Pentatax*) *thaiensis* are mussel *Pseudodon vondembuschianus ellipticus*, *Ps. cambodjensis*, *Pi. exilis*, and *Pi. e. compressa.* The hosts of *U*. (*Polyatax*) *heardi* are mussel *Hy. bialatus* and *Hy. myersiana* [7,9,10,11].

In our research, we conducted a survey of parasites in freshwater snails from the central and eastern provinces of Thailand. Remarkably, studies on water mites infesting freshwater snails from this area have been limited. Therefore, this research represents a novel report of the unionicolid water mite group found in the viviparid snail genus *Filopaludina* in Thailand.

## 2. Materials and Methods

### 2.1. Snail Specimen Collection 

A total of 39 canals and ponds in eight provinces—Bangkok, Nakhon Pathom, Nonthaburi, Samut Sakhon, Samut Prakan, and Pathum Thani for the central provinces, and Chonburi and Rayong for the eastern provinces—were designated as snail sampling sites. The freshwater snails were randomly sampled by hand using a hand net between February 2020 and March 2024 (3 times/year) and following the counts per unit of time sampling method by Olivier and Schneiderman (1956) [12]. The method involved a single collector spending approximately 20 min collecting snail samples at each sampling site. All snails collected at each sampling site were kept in plastic bags with perforation and then transported to the laboratory of the Department of Zoology, Faculty of Science, Kasetsart University, for examination of any water mite infections. Species identification of the collected snails was conducted based on their morphological features, following the identification keys of Brandt (1974) and Upatham et al. (1983) [13,14].

### 2.2. Examination of Water Mite Infection 

The snail specimens were anesthetized by immersion in 5% ethanol before being euthanized by gentle pressing between a pair of petri dishes. The pressed snail bodies were then examined under a stereomicroscope to explore the appearance of water mites in their tissue. Water mite specimens were collected using a needle and glass dropper and then washed several times in distilled water before preservation in 96% ethanol. The number of water mites and infected snails was recorded to calculate their prevalence and mean intensity, following the method of Bush et al. 1997 [15]. 

### 2.3. Morphological Study

Both fresh and preserved water mite specimens were examined and photographed for morphological characteristics using an Olympus BX51 compound light microscope with a DP70 camera (Olympus Corporation, Tokyo, Japan). The photographs of the water mites were then utilized to measure their morphological characteristics via ImageJ software version 1.54 [16]. The measured features and their methodology of measurement, modified from Vidrine (1996) [17], were as follows: TL—total body length, including the capitulum; TW—total body width, using the widest part; CL—chelicera length; LDS—length of the whole dorsal shield (both the anterior and posterior shields); LADS—length of the anterior dorsal shield; LPDS—length of the posterior dorsal shield; AC—length of the anterior coxal group (coxal plates I–II); PC—length of the posterior coxal group (coxal plates III–IV); GFL—genital field length; AGFW—width of both anterior genital field plates; AGFWS—width of a single anterior genital field plate; PGFW—width of both posterior genital field plates (present only in females); PGFWS—width of a single posterior genital field plate; GS—genital spine; P—pedipalp; DLP—dorsal length of the pedipalp segments; LI—first walking leg (leg I); LII—second walking leg (leg II); LIII—third walking leg (leg III); LIV—fourth walking leg (leg IV); TCt—trochanter (segment of a leg proximal to the coxa); BFe—basifemur (segment of a leg immediately distal to the coxa); Fe—femur (segment of the pedipalps immediately distal to the coxa); TFe—telofemur; Ge—genu (segment of the pedipalps and walking legs immediately proximal to the tibia); Ti—tibia (penultimate segment of the pedipalps and legs); and Ta—tarsus (ultimate or distal-most segment of the pedipalps and legs). The measurements were expressed in microns (µm) and as means (range: min–max). All of the type specimens were deposited in the Zoological Museum at Kasetsart University, Thailand (ZMKU). 

### 2.4. Scanning Electron Microscope (SEM) Study

The protocol for preparing well-preserved specimens of water mites for scanning electron microscopy was adapted from Chiangkul et al. (2021) [18]. Initially, all specimens were fixed in absolute ethanol for 24 h before being critically point-dried using a critical point dryer (CPD) (Quorum Technology, Lewes, UK: Polaron—model CPD7501). Subsequently, the dried specimens were mounted onto the stubs using carbon tape and coated with gold particles using a sputter coater (Quorum Technology—model SC7620). Finally, all coated specimens were examined and micrographed using a Quanta 450 scanning electron microscope at the Scientific Equipment Center, Faculty of Science, Kasetsart University.

## 3. Results and Discussion

### 3.1. Host and Infection Rate

All of the freshwater snails collected from the 39 canals and ponds in Central and Eastern Thailand between February 2020 and March 2024 were classified into 16 genera in 10 families (as seen in Table 1). Among the nineteen species examined from Bangkok and Nonthaburi provinces, two species were found to be infected with water mites: *Filopaludina martensi martensi* (Frauenfeld, 1864) and *F. sumatrensis polygramma* (Martens, 1860) (Figure 1). The first location where water mites were found was the water pond in Warunawan Urban Forest Park, Kasetsart University, Bangkok province (13°50′50.9598″ 100°33′45.8382″), followed by Klong Phra Pimol Racha, Nonthaburi province (13°55′2.8842″ 100°25′11.9274″). These two host species had different prevalences, with *F. sumatrensis polygramma* having a higher prevalence than *F. martensi martensi* (17.10% and 3.83%, respectively). However, the mean intensities of the two host species were 2.49 and 2.18 parasites/infected host (Table 2).

### 3.2. General Description

All of the collected water mites were classified as a single species in the family Unionicolidae, subfamily Unionicolinae, which is characterized by their anterior legs often having strong, movable setae; all their legs bearing simple claws; their genital fields having numerous pairs of acetabula; their females having two pairs of genital plates and heavy setae; and their palps having five segments, with P4 having prominent, pointed ventral and lateral extensions. This novel water mite species was classified as a member of the genus *Unionicola* due to its Cx III+IV being large and mostly rectangular in shape, its coxal surface having a generally reticular structure, its LI being generally stout with simple setae, and its genital field having many pairs of acetabula on its two pairs of genital plates [19]. This complex genus contains more than 45 subgenera; however, only subgenus Polyatax has been reported to parasitize freshwater mussels and viviparid snails in Asia and North America [20]. The water mites in the subgenus *Polyatax* typically have dark bodies, usually around 2.0 mm in length; pedipalps that are distinctively subcylindrical; and bifid tarsal claws midway on the dorsal side of their walking legs [17].

With characteristics of this genus exhibited in the collected water mites, particularly the shapes of their coxal plates, the characteristics of their female genital plates (two pairs of genital plates and heavy setae), the shape of their palps, and the characteristics of the setae in their leg, these specimens could be inferred to belong to the genus *Unionicola*, subgenus *Polyatax*. Additional distinctive morphological characteristics of this novel species are described as follows.
Material examinedHolotype: One mature female (ZMKU-ARA-0001) from *F. sumatrensis polygramma*, Kasetsart University, Bangkok, Thailand (13°50′50.9598″ N 100°33′45.8382″ E), 16 February 2021. Paratypes: A total of 10 individuals (6 females and 4 males) from 6 samples (4 females and 2 males) of *F. sumatrensis polygramma* (ZMKU-ARA-0002-7) and 4 samples (2 females and 2 males) of *F. martensi martensi* (ZMKU-ARA-0008-11), same locality and date as the holotype.SystematicsFamily: Unionicolidae Oudemans, 1909;Genus: *Unionicola* Haldeman, 1842; Subgenus: *Polyatax* Viets, 1933;Species: *Unionicola* (*Polyatax*) *kasetbangkhenensis* sp. nov. Diagnosis:Males (Figure 2, Figure 3, Figure 4, Figure 5 and Figure 6): The measurements based on four mature individuals; light brownish color; body oval in shape; body length, including capitulum (TL), 736 (547–900); body width (TW) 510 (304–736); chelicera length (CL) 57 (44–70); short dorsal lengths of pedipalp segments with Ge 44 (34–56), Ti 72 (52–101), and Ta 29 (26–33); setae on pedipalps with four on the femur, two on the genu, nine on the tibia, and six on the tarsus; genu of pedipalps with two long, thick setae on the anterior and posterior parts of the segment; tarsus of the pedipalp short with unequal spines; genital fields with two unfused plates on either side of the midline, with 9–10 acetabula and 3 setae (anterior, middle, and posterior) on each plate; each genital plate 74 (59–92) long (GFL) and 53 (66–131) wide (AGFWS).

First legs (LI) thick and shorter than LII–IV; LII–IV long and slender; LI with TFe 103 (80–119), Ge 133 (94–157), Ti 128 (106–151), and Ta 118 (98–156); LII with TFe 129 (88–246), Ge 140 (76–186), Ti 151 (109–180), and Ta 152 (118–208); LIII with TFe 107 (64–164), Ge 118 (63–149), Ti 116 (77–130), and Ta 115 (80–135); LIV with TFe 99 (57–131), Ge 126 (72–198), 18 (68–238), and Ta 178 (60–243); TCt cylindrical without swimming setae; long, thick setae distoventral on TFe, Ge, and Ti of LI and LIII; LIV numerous setae present on Ge and Ti; setae on legs (TCt-BFe-TFe-Ge-Ti-Ta)—LI (1–2)-3-(4–5)-(6–7)-(10–11)-9, LII 2-(5–6)-5-(8–9)-(10–11)-9, LIII (7–8)-(12–13)-(10–11)-(9–10)-(9–10)-5, and LIV (4–5)-(6–7)-(8–9)-(18–20)-(15–17)-9; claws of LI, II, and IV relatively deeply bifid with equal clawlets; and claws of LIII shallow curved bifid with equal clawlets.
Females (Figure 2, Figure 3 and Figure 7, Figure 8 and Figure 9): Measurements based on six mature individuals; dark brownish color; body oval in shape, larger than males; anterior and posterior coxal groups divided by narrow interspace; coxal plates I and II with distinct borders; coxal plates III+IV elongated (L/W 727/363 ratio 2.0); body length, including capitulum (TL), 1203 (1162–1283); body width (TW) 764 (718–837); chelicera length (CL) 124 (109–150); short dorsal length of pedipalp segments with Ge 148 (137–161), Ti 195 (110–306), and Ta 128 (106–172); setae on pedipalps with three on the femur, four on the genu, four on the tibia, and five on the tarsus; genu of pedipalps with two long, thick setae on the anterior and posterior parts of the segment; all tubercles on the tibia of pedipalps bearing thin, short setae; tarsus of pedipalps short with unequal spines; 60–66 pairs of genital acetabula lying on four genital plates; anterior genital plates with 22–26 unequal acetabula, with large inner spines on each plate; spine (GS) 91 (55–145) in length; posterior genital plates with 38–41 acetabula and some setae; genital field 250 (202–326) long (GFL), measured from anterior to posterior plates; anterior genital plate 161 (140–178) wide (AGFWS) and posterior 307 (259–348) wide (PGFWS).

First legs thick and shorter than LII–IV; LII–IV long and slender; LI with TFe 190 (131–259), Ge 310 (268–352), Ti 280 (231–333), and Ta 212 (163–252); LII with TFe 229 (169–267), Ge 282 (216–327), Ti 288 (207–333), and Ta 268 (235–315); LIII with TFe 193 (132–271), Ge 231 (185–288), Ti 211 (151–257), and Ta 199 (150–247); LIV with TFe 219 (170–269), Ge 351 (256–400), Ti 422 (268–490), and Ta 308 (206–366); trochanter cylindrical and without swimming setae; long, thick setae distoventral on the telofemur, genu, and tibia of LI and LII; LIII and LIV numerous setae present on the genu and tibia; setae on legs (TCt-BFe-TFe-Ge-Ti-Ta)—LI 3-(6–7)-6-(7–8)-(13–14)-7, LII 3-(5–6)-(9–10)-(9–10)-(11–12)-(9–11), LIII 2-(5–6)-(8–10)-(13–14)-(14–16)-(8–9), and LIV (3–4)-(4–5)-(11–13)-(30–32)-(28–29)-(14–16); claws of LI and IV relatively large bifid with equal clawlets; and claws of LII and LIII smaller than LI and LIV.

The characteristics of the setae on LIII and LIV of both sexes differ from those on LI and LII. The edges of the setae are divided into branches resembling feathers. SEM images revealed three shapes of the setae: baton-shaped, feather-shaped, and robust spine-shaped. Additionally, our observations showed periodic small thorns piled up on the claw surface of LIV (Figure 9).
Notations on the sexual differences

Female *Unionicola* (*Polyatax*) *kasetbangkhenensis* sp. nov. differ from males in that they present greater TL (1162–1283 vs. 546–900), TW (718–837 vs. 304–736), LDS (1052–1291 vs. 535), PC (555–803 vs. 271–346), GFL (202–326 vs. 59–92), AGFW (285–382 vs. 135–180), P-Ge (137–161 vs. 34–56), P-Ti (110–306 vs. 52–101), P-Ta (106–172 vs. 26–33), LI-TFe (131–259 vs. 80–119), LI-Ge (268–352 vs. 94–157), LI-Ti (231–233 vs. 106–152), LI-Ta (163–252 vs. 98–156), LIV-TFe (170–269 vs. 57–131), LIV-Ge (256–400 vs. 72–198), and LIV-Ti (268–490 vs. 68–238) and have posterior genital fields and an inner spine on their anterior genital fields.
Habitat

Type locality: *Unionicola* (*Polyatax*) *kasetbangkhenensis* sp. nov. parasitize viviparid snails (*F. sumatrensis polygramma* and *F. martensi martensi*) found in the benthic zone of ponds at Kasetsart University, main campus (Bangkhen), Bangkok, Thailand. The ponds have shallow water with a depth of approximately less than one meter, including numerous aquatic plants (duckweed, water hyacinths, creeping burhead, etc.) growing under water.
Etymology

The species epithet refers to the type locality, Kasetsart University, main campus (Bangkhen). The following common names are suggested: Kaset Bangkhen water mite (English) and Hmud Nam Kaset Bangkhen (Thai).

### 3.3. Morphological Comparison (Table 3)

Female *Unionicola* (*Polyatax*) *kasetbangkhenensis* sp. nov. differ from *Unionicola* (*Polyatax*) *heardi* Vidrine (1985) in that they present greater TL (1162–1283 vs. 900–1050), LDS (1052–1291 vs. 850–900), PC (555–803 vs. 500–600), GFL (202–326 vs. 175–200), AGFWS (285–382 vs. 260–270), P-Ge (137–161 vs. 115–130), P-Ta (105–171 vs. 40–45), LI-Ti (231–333 vs. 150–175), and LI-Ta (163–252 vs. 120–140) [7].Female *Unionicola* (*Polyatax*) *kasetbangkhenensis* sp. nov. differ from *Unionicola* (*Polyatax*) *megachela* Vidrine (1985) in that they present greater TL (1162–1283 vs. 700–800), PC (555–803 vs. 225), GFL (202–326 vs. 125–150), AGFWS (285–382 vs. 190–210), P-Ge (137–161 vs. 50–55), P-Ti (110–306 vs. 90–95), P-Ta (106–172 vs. 10–15), LI-Fe (131–259 vs. 85–95), LI-Ge (268–352 vs. 105–110), LI-Ti (231–333 vs. 100–105), LI-Ta (163–252 vs. 100–115), LIV-TFe (170–269 vs. 90–100), LIV-Ge (256–400 vs. 140–145), LIV-Ti (268–490 vs. 150–160), and LIV-Ta (206–366 vs. 150–165) [10].Female *Unionicola* (*Polyatax*) *kasetbangkhenensis* sp. nov. differ from *Unionicola* (*Polyatax*) *dobsoni* Vidrine (1985) in that they present greater TL (1162–1283 vs. 1100–1150), PC (555–803 vs. 450–500), AGFWS (285–382 vs. 240–260), P-Ge (137–161 vs. 75), and P-Ta (106–172 vs. 35–40) [10].Female *Unionicola* (*Polyatax*) *kasetbangkhenensis* sp. nov. differ from *Unionicola* (*Polyatax*) *japonensis* Viets, 1933, in that they present greater TL (1162–1283 vs. 1020), GFL (202–326 vs. 86–102), P-Ge (137–161 vs. 100), and P-Ta (106–172 vs. 62) [21].Female *Unionicola* (*Polyatax*) *kasetbangkhenensis* sp. nov. differ from *Unionicola* (*Polyatax*) *viviparaicola* Vidrine (1985) in that they present greater P-Ta (106–172 vs. 50–55) but smaller TL (1162–1283 vs. 1300–1500), LI-Fe (131–259 vs. 280–300), LI-Ge (268–352 vs. 380–420), LIV-TFe (170–269 vs. 290–310), and LIV-Ge (256–400 vs. 440–490) [10].Female *Unionicola* (*Polyatax*) *kasetbangkhenensis* sp. nov. differ from *Unionicola* (*Polyatax*) *lumbaria* Wen & Zhu (1998) in that they present greater P-Ge (137–161 vs. 62) and P-Ta (106–172 vs. 83) but smaller TW (718–837 vs. 1061), CL (109–150 vs. 211), LI-Ge (268–352 vs. 364), LI-Ta (163–252 vs. 255), LIV-TFe (170–269 vs. 273), LIV-Ti (268–490 vs. 551), and LIV-Ta (206–366 vs. 385) [21].Remarks

*Unionicola* (*Polyatax*) *kasetbangkhenensis* sp. nov. was distinguished from the other species of the subgenus *Polyatax*, including *U. heardi*, *U. megachela*, *U. dobsoni*, *U. japonensis*, *U. viviparaicola*, and *U. lumbaria*, based on the study by Vidrine (1996) and using measurements of the morphological characteristics in the females: TL, TW, CL, LDS, PC, GFL, AGFW, P-Ge, P-Ti, PTa, LI-TFe, LI-Ge, LI-Ti, LI-Ta, LIV-TFe, LIV-Ge, LIV-Ti, and LIV-Ta [16]. In addition, *U. kasetbangkhenensis* was found to parasitize freshwater snails (*F. sumatrensis polygramma* and *F. martensi martensi*), distinguishing it from other *Polyatax* in Thailand (*U. heardi*), which parasitize mussels (*Hyriopsis bialatus* Simpson, 1900).

*Unionicola* (*Polyatax*) *kasetbangkhenensis* sp. nov. is the second new species of *Polyatax* water mites reported in Thailand, following *U*. (*Polyatax*) *heardi* (Vidrine, 1985) [7,10]. Significantly, it is distinguished from other species by the pattern of the female genital field, with two pairs of genital plates and heavy setae. It also differs from other species by the structure of the tarsus segment and spines in the pedipalps and legs, the pattern of bordered medial and posterior borders on the coxal plates, the female genital field, and the pattern of claws and setae on its legs. Although this species is quite similar to *U. japonensis* in various aspects, it exhibits differences in the shape of its coxal plates III+IV, which lack straight medial margins; the fusion of male genital plates at the anterior ends; the number of acetabula on each genital plate of both males and females; and the morphology of the claws on all legs. In *U. japonensis*, the claws on legs I and IV are large and bifid, while the claws on legs II and III are small and sickle-shaped, with subequal clawlets. Additionally, *U. kasetbangkhenensis* differs from *U. heardi* in the bifid nature of the claws on all legs (LI–IV), with equal clawlets. In contrast, *U. heardi* exhibits deeply bifid claws on the first and fourth pairs of walking legs, while the claws on the second and third pairs of legs are simple and not bifid. 

Furthermore, the size of *U. kasetbangkhenensis* is similar to those of other water mites reported to be found in gastropods. However, the hosts they parasitize differ: for instance, *U. kasetbangkhenensis* feed on *F. sumatrensis polygramma* and *F. martensi martensi*, which differs from *U. heardi* in Thailand, which feed on mussels; *U. dobsoni* feed on *Campeloma geniculum* (Conrad, 1834); *U. japonensis* feed on *Cipangopaludina chinensis* (Gray, 1834), *Ci. japonica* (von Martens, 1861), and *Ci. malleata* (Reeve, 1863); *U. viviparaicola* feed on *Viviparus subpurpureus* (Say, 1829); and *U. lumbaria* feed on *Anodonta woodiana* (Lea, 1834) [7,10,17,21,22,23,24,25,26,27] (Table 3).

**Table 3 animals-14-01627-t003:** Characteristics of some species of female *Unionicola* (*Polyatax*) spp.

Characteristics (Number of Examined Specimens)	*Unionicola* (*Polyatax*) *kasetbangkhenensis* sp. nov. (6) (Female)	*Unionicola* (*Polyatax*) *heardi* Vidrine, 1985 (3)	*Unionicola* (*Polyatax*) *megachela* Vidrine, 1985 (3)	*Unionicola* (*Polyatax*) *dobsoni* Vidrine, 1985 (2)	*Unionicola* (*Polyatax*) *japonensis* Viets, 1933 (1)	*Unionicola* (*Polyatax*) *viviparaicola* Vidrine, 1985 (3)	*Unionicola* (*Polyatax*) *lumbaria* Wen & Zhu, 1998 (4)
Total length (TL)	1203 (1162–1283)	963 (900–1050)	767 (700–800)	1125 (1100–1150)	1020	1417 (1300–1500)	1190
Total width (TW)	764 (718–837)	n/a	n/a	n/a	n/a	n/a	1061
Chelicera length (CL)	124 (109–150)	n/a	n/a	n/a	n/a	n/a	211
Length of dorsal shield (LDS)	1171 (1052–1291)	883 (850–900)	n/a	n/a	n/a	n/a	n/a
Length of posterior coxal group (PC)	693 (555–803)	558 (500–600)	225	475 (450–500)	462	773 (750–820)	n/a
Genital field long (GFL)	250 (202–326)	188 (175–200)	135 (125–150)	195 (160–230)	86–102	280	n/a
Anterior genital field wide (AGFW)	33 (285–387)	265 (260–270)	200 (190–210)	250 (240–260)	n/a	350	296
Anterior genital field wide—one side (AGFWS)	161 (140–1178)	n/a	n/a	n/a	n/a	n/a	n/a
Posterior genital field wide (PGFW)	307 (259–348)	n/a	n/a	n/a	n/a	n/a	n/a
Posterior genital field wide—one side (PGFWS)	136 (112–154)	n/a	n/a	n/a	n/a	n/a	n/a
Number of genital acetabula/side	60–66	16–23	5	8	9–12	18–30	5
Dorsal lengths of pedipalp segments							
P3 = Ge	148 (137–161)	63 (60–65)	53 (50–55)	75	100	n/a	62
P4 = Ti	195 (110–306)	123 (115–130)	94 (90–95)	168 (165–170)	162	185 (180–190)	144
P5 = Ta	128 (106–172)	41 (40–45)	13 (10–15)	38 (35–40)	62	53 (50–55)	83
Dorsal lengths of leg segments: leg I							
P3 = TFe	190 (131–259)	158 (150–175)	90 (85–95)	228 (225–230)	198	285 (280–300)	229
P4 = Ge	310 (268–352)	203 (180–230)	108 (105–110)	330 (320–340)	270	395 (380–420)	364
P5 = Ti	280 (231–333)	164 (150–175)	102 (100–105)	263 (250–275)	244	292 (280–300)	309
P6 = Ta	212 (163–252)	130 (120–140)	107 (100–115)	215 (210–220)	231	237 (230–240)	255
Dorsal lengths of leg segments: leg IV							
P3 = TFe	219 (170–269)	179 (165–195)	93 (90–100)	233 (230–235)	207	300 (290–310)	273
P4 = Ge	351 (256–400)	278 (255–300)	142 (140–145)	363 (350–375)	310	460 (440–490)	385
P5 = Ti	422 (268–490)	320 (295–350)	157 (150–160)	383 (375–390)	350	463 (440–490)	551
P6 = Ta	308 (206–366)	239 (230–250)	155 (150–165)	313 (310–315)	317	365 (350–375)	385
Distributions	Thailand	Thailand	Louisiana, USA	Florida, USA	Asia (Japan) and USA	Louisiana, USA	China
Hosts	Snail (*F. sumatrensis polygramma* and *F. martensi martensi*)	Mussel (*Hyriopsis bialatus* Simpson, 1900)	Mussel (*Amblema dombeyana* (Valenciennes, 1827))	Snail (*Campeloma geniculum* (Conrad, 1834))	Snail (*Cipangopaludina chinensis* (Gray, 1834), *Ci. japonica* (von Martens, 1861), and *Ci. malleata* (Reeve, 1863))	Snail (*Viviparus subpurpureus* (Say, 1829)	Mussel (*Anodonta woodiana* (Lea, 1834)
References	This study	[7]	[10]	[10]	[21]	[10]	[22]

## 4. Conclusions

This discovery of *Unionicola* (*Polyatax*) *kasetbangkhenensis* adds a new member to the subgenus *Polyatax* from Thailand, and this species is the second of *U*. (*Polyatax*) in Thailand after Vidrine (1985) found *U*. (*Polyatax*) *heardi* in 1985.

## Figures and Tables

**Figure 1 animals-14-01627-f001:**
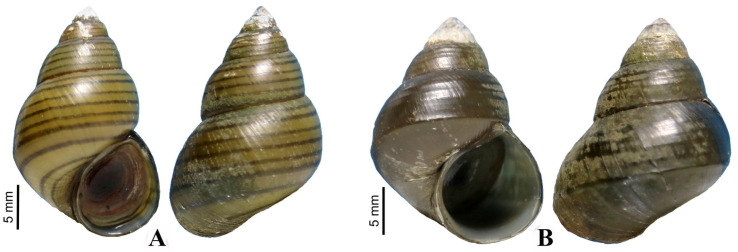
*Filopaludina sumatrensis polygramma* (**A**) and *Filopaludina martensi martensi* (**B**).

**Figure 2 animals-14-01627-f002:**
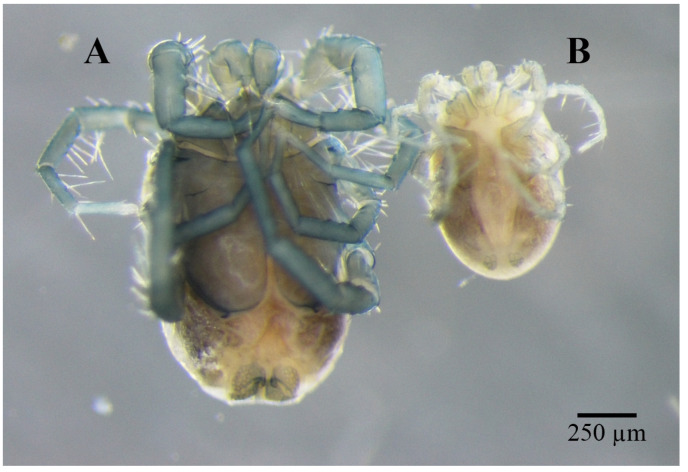
Female (**A**) and male (**B**) *Unionicola* (*Polyatax*) *kasetbangkhenensis* sp. nov.

**Figure 3 animals-14-01627-f003:**
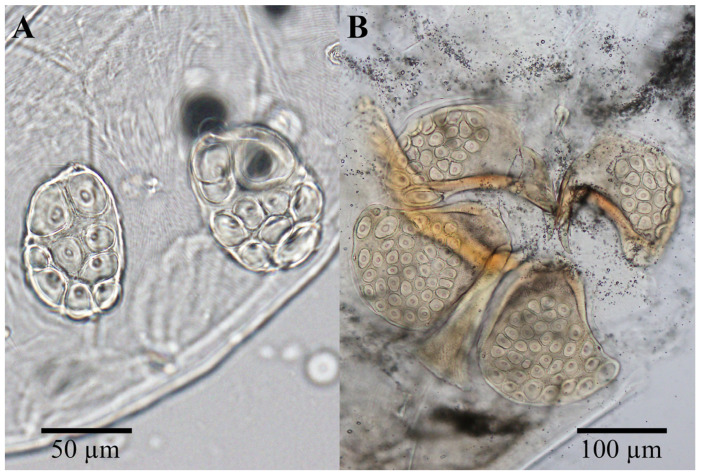
Genital fields of male (**A**) and female (**B**) of *Unionicola* (*Polyatax*) *kasetbangkhenensis* sp. nov.

**Figure 4 animals-14-01627-f004:**
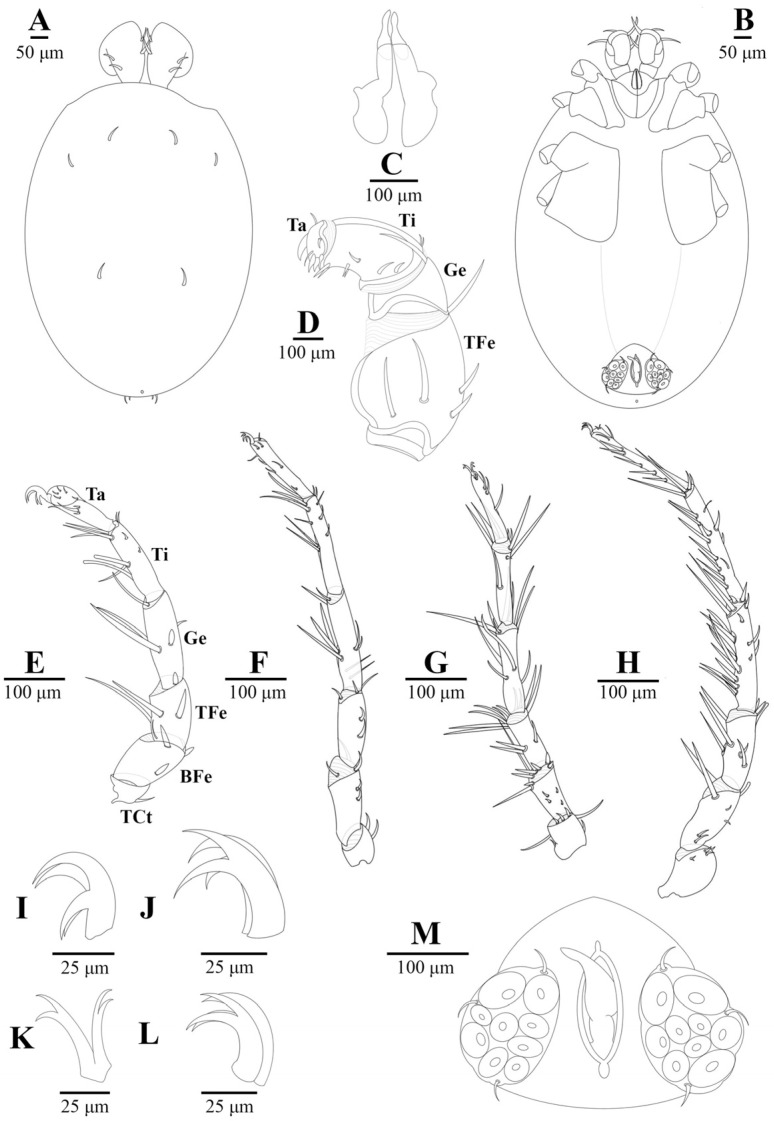
*Unionicola* (*Polyatax*) *kasetbangkhenensis* sp. nov., male: (**A**) dorsal view; (**B**) ventral view; (**C**) chelicera; (**D**) pedipalp; (**E**) leg 1; (**F**) leg 2; (**G**) leg 3; (**H**) leg 4; (**I**) claw of leg 1; (**J**) claw of leg 2; (**K**) claw of leg 3; (**L**) claw of leg 4; (**M**) genital plates.

**Figure 5 animals-14-01627-f005:**
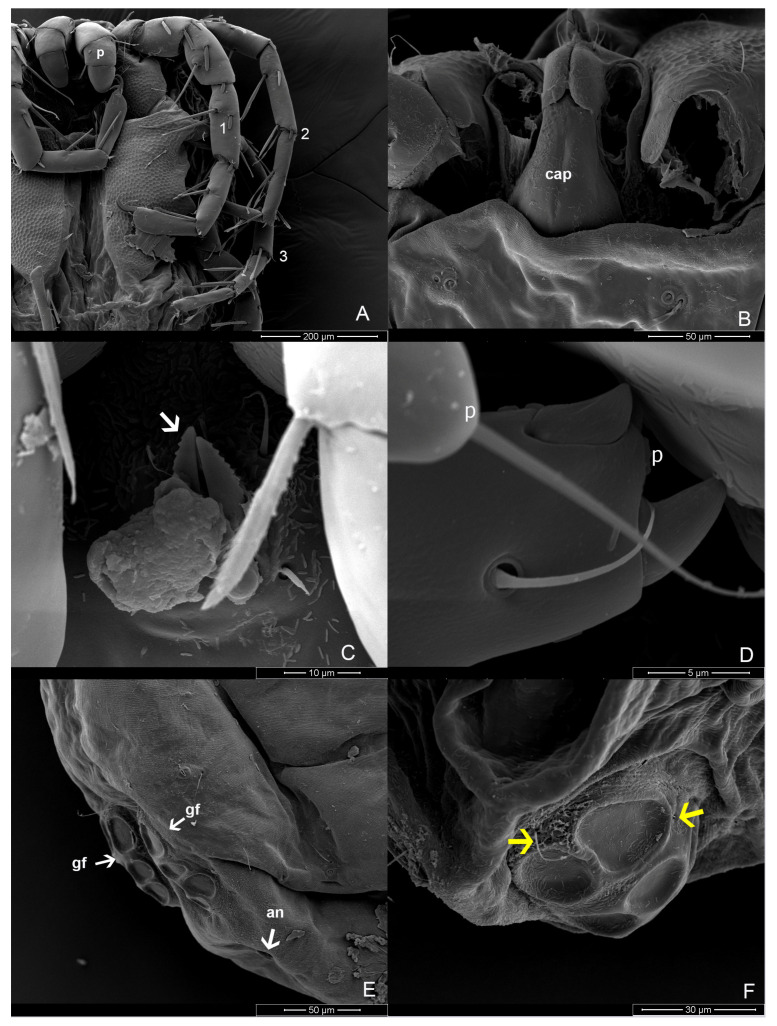
*Unionicola* (*Polyatax*) *kasetbangkhenensis* sp. nov. male: (**A**) ventral view showing pedipalp (p) on leg 1 (1), leg 2 (2), and leg 3 (3); (**B**) dorsal view showing capitulum (cap); (**C**) ventral view showing chelicerae (white arrow); (**D**) ventral view showing tarsus of pedipalp; (**E**) ventral view showing genital field (gf) and anus (an); (**F**) ventral view showing filament spine (yellow arrows).

**Figure 6 animals-14-01627-f006:**
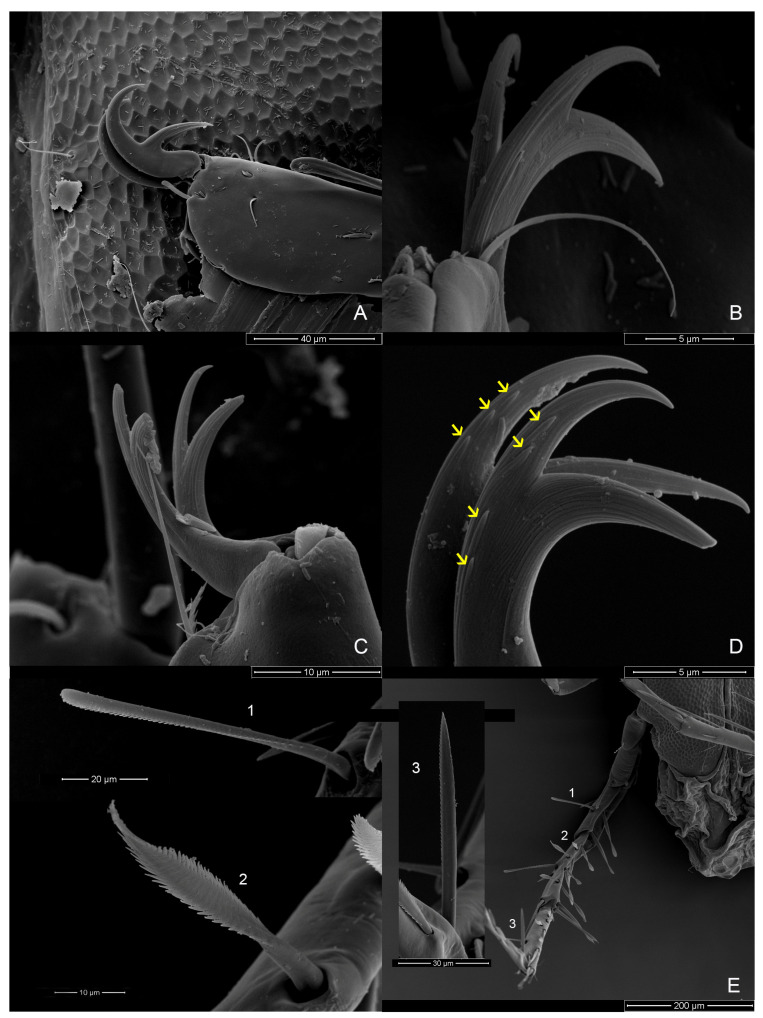
*Unionicola* (*Polyatax*) *kasetbangkhenensis* sp. nov., male, ventral view: (**A**) leg 1; (**B**) claw of leg 2; (**C**) claw of leg 3; (**D**) claw of leg 4, showing a little spine on the claw (yellow arrow); (**E**) leg 4, showing 3 types of spines: 1 = baton shape, 2 = feather-like, 3 = strong spine.

**Figure 7 animals-14-01627-f007:**
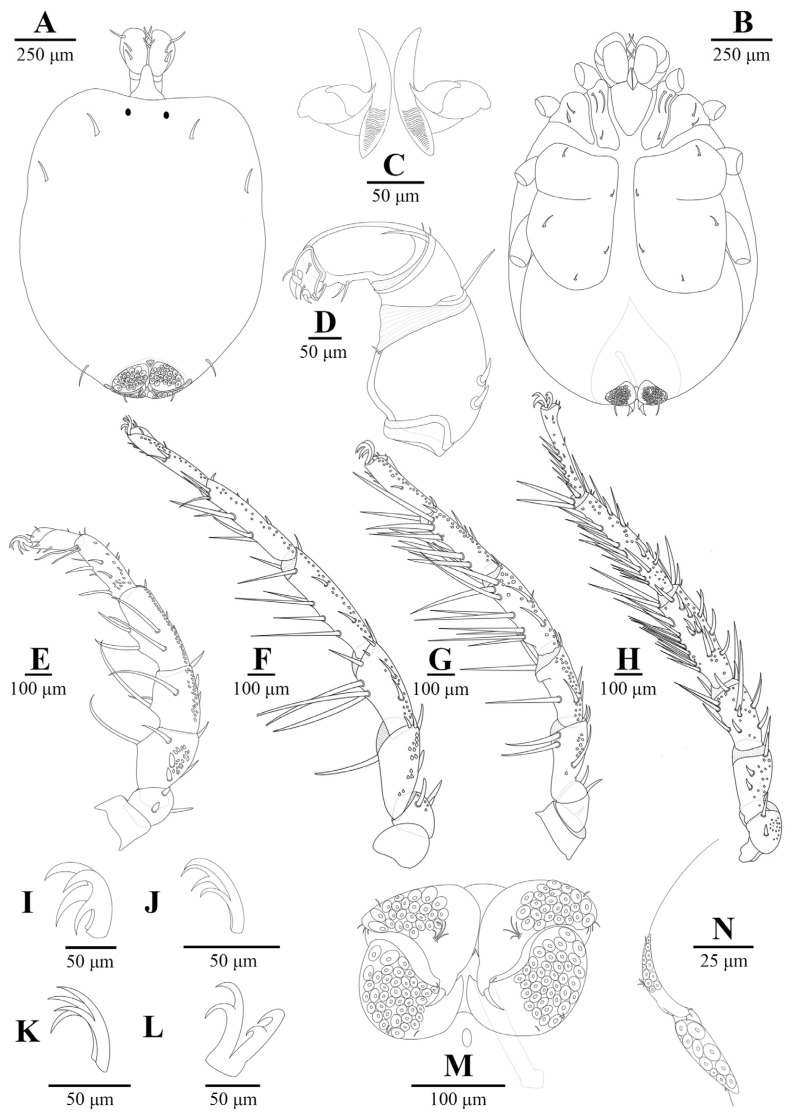
*Unionicola* (*Polyatax*) *kasetbangkhenensis* sp. nov., female: (**A**) dorsal view; (**B**) ventral view; (**C**) chelicera; (**D**) pedipalp; (**E**) leg 1; (**F**) leg 2; (**G**) leg 3; (**H**) leg 4; (**I**) claw of leg 1; (**J**) claw of leg 2; (**K**) claw of leg 3; (**L**) claw of leg 4; (**M**) genital plates; (**N**) side view of genital plates.

**Figure 8 animals-14-01627-f008:**
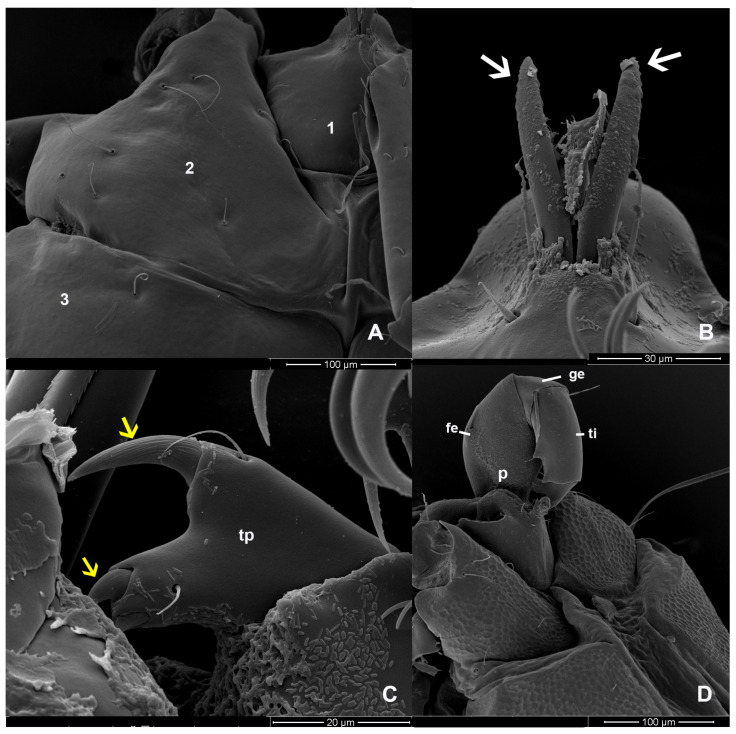
*Unionicola* (*Polyatax*) *kasetbangkhenensis* sp. nov., female, ventral view: (**A**) coxal plates 1, 2, and 3; (**B**) chelicerae (white arrow); (**C**) tarsus of pedipalp (tp) with 2 strong spines (yellow arrow); (**D**) pedipalp (p) showing segments of the pedipalp: femur (fe), genu (ge), and tibia (ti).

**Figure 9 animals-14-01627-f009:**
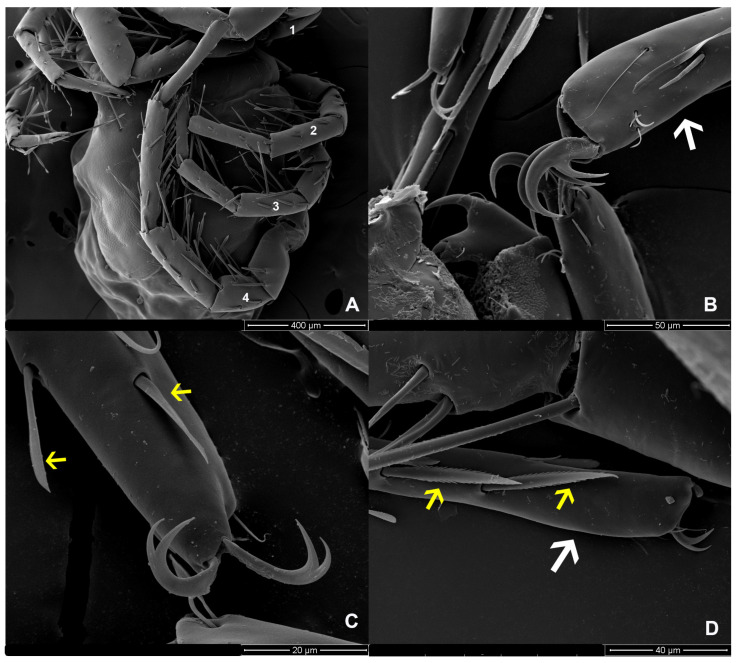
*Unionicola* (*Polyatax*) *kasetbangkhenensis* sp. nov., female leg, ventral view: (**A**) legs 1, 2, 3, and 4; (**B**) leg 1 (white arrow); (**C**) tarsus of leg 2 with blade-like spines (yellow arrow); (**D**) tarsus of leg 3 (white arrow) with spines in which the edges are streaked like feathers (yellow arrow).

**Table 1 animals-14-01627-t001:** List of freshwater snails found in central and eastern Thailand in this study.

Families	Genus	Species
Ampullariidae	*Pomacea*	*Po. canaliculata* (Lamarck, 1819)
Assimineidae	*Cyclotropis*	*Cy. carinata* (Lea, 1856)
Bithyniidae	*Bithynia*	*B. siamensis siamensis* (Lea 1856)
Lymnaeidae	*Austropeplea*	*Au. viridis* (Quoy & Gaimard, 1832)
	*Radix*	*R. auricularia* (Linnaeus, 1758)
Nassariidae	*Anentome*	*An. helena* (von dem Busch, 1847)
Pachychilidae	*Sulcospira*	*Su. housei* (Lea, 1856)
Physidae	*Physella*	*Ph. acuta* (Draparnaud, 1805)
Planorbidae	*Gyraulus*	*G. siamensis* (von Martens, 1867)
	*Indoplanorbis*	*In. exustus* (Deshayes, 1834)
Thiaridae	*Melanoides*	*M. tuberculata* (Müller, 1774)
	*Sermyla*	*Se. riqueti* (Grateloup, 1840)
	*Tarebia*	*Ta. granifera* (Lamarck, 1822)
	*Thiara*	*Th. scabra* (Müller, 1774)
Viviparidae	*Filopaludina*	*F. martensi cambodiensis* (Brandt, 1974)
		*F. martensi martensi* (Frauenfeld, 1864)
		*F. sumatrensis polygramma* (Martens, 1860)
		*F. sumatrensis speciosa* (Deshayes, 1876)
	*Idiopoma*	*Id. umbilicata* (Lea, 1856)

**Table 2 animals-14-01627-t002:** Prevalence and mean intensity of water mite infection in freshwater snail *Filopaludina* spp.

	No. Examined	No. Infected	Prevalence (%)	Mean Intensity (Parasites/Infected Snail)
*Filopaludina sumatrensis polygramma*	310	53	17.10	2.49
*Filopaludina martensi martensi*	287	11	3.83	2.18

## Data Availability

Data are available in The Official Registry of Zoological Nomenclature—Zoobank: urn:lsid:zoobank.org:act:F3F6A2C2-A40B-4307-83D1-281847AAEF40.

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
