# Peer review of "The First Report of a Water Mite Unionicola (Trombidiformes: Unionicolidae) Infection in Filopaludina spp. (Gastropoda: Viviparidae) from Thailand with a Description of a New Species of Unionicola (Polyatax) kasetbangkhenensis sp. nov.â€"

_animals, 2024, doi:10.3390/ani14111627_

Round 1
Reviewer 1 Report
Comments and Suggestions for Authors
The manuscript "First report of water mite Unionicola (Trombidiformes: 2 Unionicolidae) infection in Filopaludina spp. (Gastropoda: 3 Viviparidae) from Thailand with the description of new species 4 of Unionicola (Polyatax) kasetbangkhenensis sp. nov.” is well prepared with good drawings and photos, and worthy of publication in Animals.
In general comments I have two suggesions:
1. The name of the species derived from locus typicus is a very good option, but I suggests to made it shorter and easier to pronounce – but it is only suggestion.
2. The key for identification will be useful.
I have some detailed comments:
Lines: 20-21 and 45-46; Water mites belong to Parasitengona that maens larvae are almost always parasites on aquatic insects, but adult and deutonymps are predators, excluding unionicolis which are parasites. But the authors sentences suggest that adults are predators and parasites – should be changed.
Lines: 69-70; „The freshwater snails were randomly sampled by hand and hand net 69 between February 2020 and March 2024,” – authors should explain in exactly months snails were sampled – every months.
Lines: 126-132; in which months infected snails were collected? Were there differences between prevalence in particular months?
Lines: 145-149; There are features that better characterize both Unionicolidae and Unionicola, the authors should mention them – see: „Smit, H. Water mites of the world, with keys to the families, subfamilies, genera and subgenera (Acari: Hydrachnidia); Nederlandse 424 Entomologische Vereniging. 2020, pp. 1-774.” or „Gerecke, R., Gledhill, T., Pešić, V. & Smit, H. (2016) Chelicerata: Acari III. In: Gerecke, R. (Ed.), Süßwasserfauna von Mitteleuropa, 7/2-3, Springer-Verlag Berlin, Heidelberg, pp. 1–429.”
Lines: 370-371; „with anterior acetabular plates each bearing two acetabula and a short thick spine on the inner margin” – I do not understand, I can see many acetabulas
Author Response
We appreciate the reviewer for your precious time in reviewing our paper and providing valuable comments. I already edited and responded to your comments as shown in this attached file.
Best regards,

Reviewer 2 Report
Comments and Suggestions for Authors
General comments:
This paper contains a description of new water mite species in Thailand. However, several statements will require further justification.
This paper is highly descriptive. Description of a new water mite species is worth to be published in Animals. However, the point of an argument on discussion was not clearly given. Therefore, I did not understand what the point of issue was and what the leading solution for the issue was. The results of the present study were not adequate for making discussion. If the authors want to just describe the newly obtained information on the water mite species in Thailand, just a description of the new species will be presented without discussion.
Specific comments are directly given in “animals-3010673-peer-review-v1”.

I strongly suggest that the English should be checked by native English speakers.
Author Response

(The authors gave the same response as above.)

Round 2
Reviewer 2 Report
Comments and Suggestions for Authors
General comments:
This paper contains interesting information related to a new water mite species in Thailand. The manuscript has been revised well. I think this manuscript will be acceptable after some minor revisions have been done.
Specific comments:
Abbreviations are used in the text/body, but not in the figure legends. Is it not necessary to unify the terminology throughout the document in this journal?
Generally, an endash (not hyphen) is used to represent a range. Is it not necessary to unify this format throughout the document in this journal?
i.e.,
736 (547-900), LI (1-2)-3-(4-5)-(6-7)-(10-11)-9
736 (547–900), LI (1–2)-3-(4–5)-(6–7)-(10–11)-9
P2. Lines 64–65.
Introduction could not contain the present results. “Our findings revealed that only two gastropod species in the family Viviparidae were infested by the water mites.” should be deleted. Or replace this sentence with that in another published paper.
P.10, Line 269: “Notations on the sexual differences from an inter-specific comparison” will be changed into “Notations on the sexual differences”.
Author Response
Thank you very much for your valuable comments. We have made the changes as shown in the file "response to reviewer2" as attach
